# The Influence of Age on the Wood Properties of *Paulownia tomentosa* (Thunb.) Steud.

Bruno Esteves [1,2,*], Luísa Cruz-Lopes [2,3], Hélder Viana [4,5], José Ferreira [1,2], Idalina Domingos [1,2] and Leonel J. R. Nunes [6,7]

1 Department of Wood Engineering, Polytechnic Institute of Viseu, Av. Cor. José Maria Vale de Andrade, 3504-510 Viseu, Portugal; jvf@estgv.ipv.pt (J.F.); ijd@estgv.ipv.pt (I.D.)
2 Centre for Natural Resources, Environment and Society-CERNAS-IPV Research Centre, Av. Cor. José Maria Vale de Andrade, 3504-510 Viseu, Portugal; lvalente@estgv.ipv.pt
3 Department of Environmental Engineering, Polytechnic Institute of Viseu, Av. Cor. José Maria Vale de Andrade, 3504-510 Viseu, Portugal
4 Department of Ecology and Sustainable Agriculture, Polytechnic Institute of Viseu, Av. Cor. José Maria Vale de Andrade, 3504-510 Viseu, Portugal; hviana@esav.ipv.pt
5 CITAB, Centre for the Research and Technology of Agro-Environmental and Biological Sciences, University of Trás-os-Montes and Alto Douro, Quinta de Prados, 5000-801 Vila Real, Portugal
6 PROMETHEUS, Unidade de Investigação em Materiais, Energia e Ambiente para a Sustentabilidade, Escola Superior Agrária, Instituto Politécnico de Viana do Castelo, Rua da Escola Industrial e Comercial de Nun'Alvares, 4900-347 Viana do Castelo, Portugal; leonelnunes@esa.ipvc.pt
7 CEF, Centro de Estudos Florestais, Instituto Superior de Agronomia, Universidade de Lisboa, Tapada da Ajuda, 1349-017 Lisboa, Portugal
* Correspondence: bruno@estgv.ipv.pt

**Abstract:** Forests in Portugal are highly dependent on a short number of wood species, and new species with higher profitability are needed. The *Paulownia* species has generated great interest due to its fast-growing and relatively good wood properties. However, environmental factors have shown that *Paulownia* grows differently in each case. This study intends to determine the properties of young Paulownia trees from Portuguese plantations to determine the best age to cut the trees according to their use. The chemical composition (extractives in dichloromethane, ethanol and water, lignin, $\alpha$-cellulose, and hemicelluloses), heating value, elemental analysis (CHNO), inorganic elements and thermogravimetric analyses (TGA), and the most important mechanical and physical properties (density, MOE, bending strength, water absorption equilibrium, moisture content, and dimensional changes) were determined for 1-, 3-, and 5-year-old trees. The results show that, chemically, the extractives increased while hemicelluloses decreased with age, and no trend was found for lignin and $\alpha$-cellulose. The physical and mechanical properties increased with age, except for the MOE and bending strength. The 5-year-old samples presented the best features for pellet production, namely, calorific power, elemental composition CHNO and sulfur, and inorganic elements.

**Keywords:** age; chemical composition; elemental composition; mechanical properties; *Paulownia tomentosa*; physical properties

## 1. Introduction

The introduction of new species in forest management has gained attention regarding the prevention of the spread of invasive species. However, new species with higher profitability are needed to increase the value of forest products and the resilience of rural populations. The genus *Paulownia*, belonging to the family *Paulowniaceae*, consists of nine species: *Paulownia albiphloea*, *Paulownia australis*, *Paulownia catalpifolia*, *Paulownia elongata*, *Paulownia fargesii*, *Paulownia fortunei*, *Paulownia kawakamii*, *Paulownia taiwaniana*, and *Paulownia tomentosa* [1], and also by several hybrids, such as Cotevisa 2, produced in Spain, and which is the result of the cross between *P. elongata* x *P. fortunei*.

*Paulownia tomentosa* (Thunb.) Steud. is a deciduous tree that, in natural conditions, reaches heights of 20–30 m [2,3], while its diameter can reach 2 m [2]. Usually, this plant tends to form many branches if it is grown in an open space, but if it has to compete with other species, it tends to form a straight trunk [3]. *P. tomentosa* is a species that has generated great interest around the world, and is referred to as the miracle tree. This classification is due to this species being a fast-growing tree with a high rate of carbon absorption and showing good fire resistance. The high growth rate allows exploitation in short rotation periods, rather than waiting for dozens of years, avoiding the risk of fires destroying years of hard labor.

The wood of *P. tomentosa* is, according to Icka et al. [3], lightweight, sturdy, quick drying, and easy to shape, which makes it suitable for sculptures and insulation [4,5]. This species has a specific mass of approximately 0.35 $g \cdot cm^{-3}$ [4], a low thermal conductivity $(0.063–0.086 \ kcal \cdot m^{-1} \cdot h^{-1} \cdot °C^{-1})$, natural fire resistance [6], and durability against several xylophages because of its high tannin content [3]. Paulownia leaves a have high protein content, so that they can be used for animal fodder. This species is still considered environmentally friendly because it has a high carbon fixation coefficient [3], as the tree absorbs about 22 kg of $CO_2$ and produces 6 kg of $O_2$ per year [3].

The productivity of *Paulownia* spp. varies according to several factors (e.g., species, site of plantation, climatic conditions, density, silvicultural management, and irrigation) in different regions of the world. The genus is very adaptable, extremely fast growing, and provides a great potential for extensive cultivation. In China, more precisely in Juan Cheng County, in the Shantung Province, under normal edaphoclimatic conditions, where the genus is native, 10-year-old *P. tomentosa* DBH (diameter at breast height) reaches 28.1 cm, and *P. elongate* DBH reaches 39.6 cm, with an average timber volume of 0.24 and 0.62 $m^3$, respectively [7]. Barton et al. [8] reported 11 to 30 $m^3 \cdot ha^{-1} \cdot year^{-1}$ in Brazil, and 6.85 $m^3 \cdot ha^{-1} \cdot year^{-1}$ from 40-year-old Paulownia plantations in the USA. For the biomass production, accumulated productivities in stems can reach up to 69 $ton \cdot ha^{-1}$ for three-year cycles [9]. In a semi-arid Mediterranean environment (Spain), García et al. [10], in an experiment with hybrids (*Paulownia elongate* x *fortune*), reported a productivity of 6 $ton \cdot ha^{-1}$ of total biomass, in the best conditions, in stands of 17 months. Zuazo et al. [11] studied a plantation with 24-month-old trees, using the clones Cotevisa 2 and Suntzu 11, with a productivity of 7.2 and 14.0 $ton \cdot ha^{-1}$, respectively.

The main objective of the present research is to determine the properties of the species *Paulownia tomentosa* from specimens planted in Portugal, to assess the best possibilities for the recovery and valorization of the wood obtained from specimens aged 1, 3, and 5 years old, respectively.

## 2. Materials and Methods

### 2.1. Sampling and Material Preparation

Young Paulownia wood samples that are 1, 3, and 5 years old from a plantation in the Viseu region, Portugal, were used for the present study. The number of trees were 6, 2, and 2, respectively. The preparation of the samples started with them being cut into pieces of around 1 $cm^3$ with a chisel and a hammer, excluding all the bark. After drying the samples naturally in open air and at room temperature, they were crushed in the Retsch SMI mill. Subsequently, sifting was performed for half an hour in a vibratory sieve model Retsch 5657 HAAN 1, and the following fractions were obtained: >40 mesh (>0.425 mm), 40–60 mesh (0.425–0.250 mm), 60–80 mesh (0.250–0.180 mm), and 80 mesh (<0.180 mm) for 30 min.

#### 2.1.1. Ash Content

To determine the ash content for the barks and branches of *P. tomentosa*, the TAPPI T 211 om-93 standard was followed. For this, 20 g ($\pm$0.0001 g) of a 40-mesh fraction sample

was used and incinerated in a muffle at 525 °C for 3 h. The ash content (Z) was determined by Equation (1).

$$Z\ (\%) = \left( \frac{ash\ mass}{dry\ sample\ mass} \right) \times 100 \tag{1}$$

### 2.1.2. Extractive Content

The extractive content was determined by sequential solvent extraction. For this purpose, 10 g (±0.0001 g) of a sample (of 40–60 mesh fraction) was used in a Soxhlet apparatus and extracted with solvents of increasing polarity: dichloromethane (6 h), ethanol (16 h), and hot water (16 h). The extractive content (E) was determined by Equation (2).

$$E\ (\%) = \left( \frac{mass\ of\ extractives}{dry\ sample\ mass} \right) \times 100 \tag{2}$$

### 2.1.3. Lignin Content

For the determination of the lignin content by the Klason method, about 350 mg of each sample were used, from a sample free of extractives, as described in the Standard TAPPI T222 om-02 [12], which quantifies lignin as a solid residue. This method is based on two hydrolyses. The first is with sulfuric acid at 72% for 1 h at 30 °C, and the second is with sulfuric acid at 3% for 4 h in reflux. Since this procedure is time consuming, the second hydrolysis was replaced by autoclave hydrolysis at 120 °C for 1 h. After hydrolysis, the samples were filtered in pre-heavy nr. 4 crucibles, washed with warm water, and dried in an oven at 60 °C overnight, followed by 1 h at 100 °C. The lignin content (L) was calculated using Equation (3).

$$L\ (\%) = \left( \frac{lignin\ mass}{dry\ sample\ mass} \right) \times 100 \tag{3}$$

### 2.1.4. Holocellulose Content

The holocellulose content was determined for the extracted wood by the acid chlorite method, in which holocellulose is obtained as an insoluble residue. The method uses 2 g of extractive free wood placed in a 1 L flask with 160 mL of distilled water at 70 °C, 20 mL of a solution with 8.5 g of sodium chlorite in 250 mL of distilled water, and other with 13.5 g of NaOH in 50 mL of distilled water, and 37.5 of acetic acid. A total of 20 mL of each solution was added until the sample became white, which can take up to 8 h [13]. The samples were then filtered in a nr. 2 crucible and washed with cold water, and 15 mL of acetone. The holocellulose (HC) content was determined by Equation (4).

$$HC\ (\%) = \left( \frac{holocellulose\ mass}{dry\ sample\ mass} \right) \times 100 \tag{4}$$

### 2.1.5. $\alpha$-Cellulose Content

The $\alpha$-cellulose content was determined for the dry holocellulose by two subsequent hydrolyses with 17.5% NaOH at 20 °C for 30 min and 15 min of rest outside the bath. Afterwards, 8.25 mL of water were added to the samples that remained for 1 additional hour at 20 °C in the thermal bath. Then, the samples were filtered and washed using 25 mL of NaOH at 8.3% and distilled water, ending with 3.75 mL of 10% acetic acid. The samples were dried at 105 °C overnight and the $\alpha$-cellulose content (C) was determined by the following Equation (5):

$$C\ (\%) = \left( \frac{cellulose\ mass}{dry\ sample\ mass} \right) \times 100 \tag{5}$$

### 2.1.6. Hemicellulose Content

The hemicellulose content was determined indirectly by the difference between holocellulose and $\alpha$-cellulose, using Equation (6).

$$\text{Hemicellulose (\%)} = \text{HC (\%)} - \text{C (\%)} \tag{6}$$

### 2.1.7. Chlorine Determination, Heating Value, Elemental Analysis (CHNO), and Thermogravimetric Analysis (TGA)

To carry out the laboratory characterization tests, the procedures presented and referred to in the ENPlus® standard were used. The choice to use this type of reference was made because this standard allows for a comparison with the results obtained via other species that are used for more general purposes, for example, to produce bioenergy. Thus, laboratory characterization tests were performed using the following standards:

○ ISO 17225-1: 2014—Solid biofuels—Fuel specifications and classes—Part 1: General requirements;
○ ISO 16948: 2015—Solid biofuels—Determination of total content of C, H, and N;
○ ISO 16967: 2015—Solid biofuels—Determination of major elements—Al, Ca, Fe, Mg, P, K, Si, Na, and Ti;
○ ISO 16968: 2015—Solid biofuels—Determination of minor elements—Ar, Cd, Co, Cr, Cu, Hg, Mn, Mo, Ni, Pb, Sb, V, and Zn;
○ ISO 16994: 2016—Solid biofuels—Determination of total content of S and Cl;
○ ISO 18125: 2017—Solid biofuels—Determination of heating value.

### 2.1.8. Physical and Mechanical Properties

The density was determined for wood conditioned at 20 °C and 65% relative humidity by weighing a cubic sample with a 20 mm edge and measuring the wood dimensions in the three directions. An average of 10 replicates was used. The bending strength and stiffness were determined in wooden test specimens by a three-point bending test in a Servosis ME-405/5 universal test machine with 360 mm × 20 mm × 20 mm in transverse, radial, and tangential directions, respectively, in accordance with the Portuguese Standard NP-619 [14]. It was not possible to cut well-oriented 1-year-old Paulownia samples for bending strength due to the small size of the trees. The samples were conditioned at 20 °C and 65% relative humidity prior to testing. Ten replicates were made for each assay. The specimens were placed with the radial face oriented upwards and supported on two supports at a distance of 300 mm. The support was moved until the pulley was against the specimen without exerting any force. The test was performed at a constant speed of 3 mm·min$^{-1}$. The same speed was maintained until a force of 735.5 N was reached in the load cell, at which time the test was ended. The force 735.5 N was 10% of the average maximum force obtained from the bending strength assays. The modulus of elasticity (MOE) was calculated according to Equation (7)

$$\text{MOE(MPa)} = \frac{\Delta F \times L^3}{\Delta x \times h \times h^3} \times 9.8 \tag{7}$$

where $\Delta F/\Delta x$ corresponds to the slope of the elastic zone in kg·mm$^{-1}$, L is the length of the span between the two axes in mm, h is the height, and b the width of the specimen, both expressed in mm. The test used to determine the bending strength was performed on the same machine used to determine the modulus of elasticity. The specimens were placed as mentioned above. The average speed of the assay was calculated so that rupture occurred approximately 2 min after the start of the assay. The bending strength was calculated according to Equation (8):

$$\text{Bending strength (MPa)} = \frac{3 \times L}{2 \times b \times h^2} \times 9.8 \tag{8}$$

Ten replicates were made for each sample. The equilibrium moisture content and water absorption were determined in wood cubic samples, with approximately 20 mm edges using 3 cycles of 0% EMC (oven at 100 °C) and 100% (water at 20 °C). Swelling was determined as the difference between the dimensions of the initial dry wood and the dimensions of the samples in 0% and 100% moisture environments for the 3 cycles.

## 3. Results and Discussion

The wood chemical composition is the basis for studying the possible uses of Paulownia wood. Figure 1 presents the chemical composition of Paulownia wood at 1, 3, and 5 years old. The standard deviations are presented as error bars. The results show that, generally, there is an increase in the extractive content with the age of the tree and that this increase is mainly due to ethanol and water extracts. Similar results were obtained by Domingos et al. [15], which stated that the extractive content of eucalyptus wood significantly increases with the ages of trees ranging from 6 up to 15 years, especially regarding ethanol extractives. Likewise, Miranda and Pereira [16] determined the variation of the chemical composition for *Eucalyptus globulus* trees that were 2, 3, and 6 years old, and found that there was an increase in the extractive content, which are mainly soluble in water and ethanol in to tree age. These extracts are generally composed of phenolic compounds, such as tannins, lignans, stilbenes, short-chain phenols, phytosterols, and carbohydrates. *Paulownia tomentosa* wood in Brazil, with growth that is 13 years old, has shown a higher extractive content (17.1%), which might mean that the amount of extractives will still increase after 5 years [17]. Non-polar extractives generally represent a small percentage of extractives in fast-growing trees, such, as for instance, poplars [18,19]

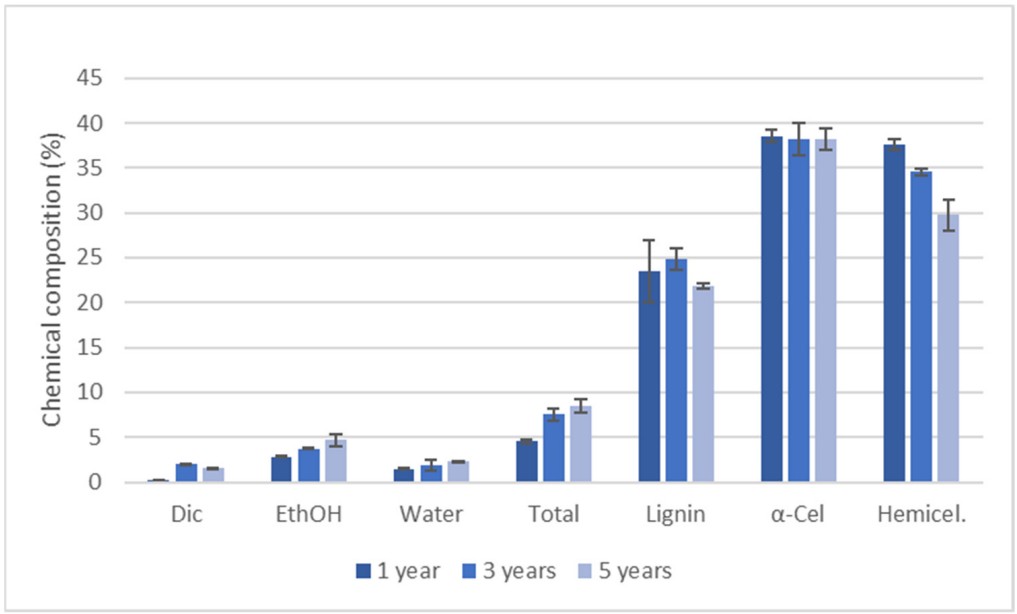

**Figure 1.** Chemical composition of Paulownia wood that is 1, 3, and 5 years old.

There seems to be no substantial differences between α-cellulose in samples, until they reach 5 years of age. On the other hand, there is a decrease in the hemicellulose content with the progression of age. The results previously presented for 13-year-old *Paulownia tomentosa* wood growth in Brazil showed a much higher α-cellulose content (62%) and much lower hemicellulose content (5%). No trends for hemicellulose were found by Miranda and Pereira [16] for eucalyptus wood.

In relation to the lignin content, the increase verified between the 1- and 3-year-old trees was followed by a decrease in the 5-year-old samples, but the differences were relatively small. The same was observed by Domingos et al. [15] for eucalyptus wood, for which the percentage of Klason Lignin was approximately constant between wood that was 6 to 15 years old, ranging from 19.6% to 23.1%. On the other hand, according to Miranda and Pereira [16], there was an increase in the lignin content of *Eucalyptus globulus* trees that were 2, 3, and 6 years old. The same was found concerning soluble lignin content, which varied between 3.3% and 4.0%. Similar results were reported for *Eucalyptus grandis* woods of four different ages (10, 14, 20, and 25 years) [20].

The knowledge of the variation of chemical compositions with age allows us to choose the ideal cutting age according to the use designated to the wood. As Paulownia is a rapid-growth species, the possibility of using wood in just a few years compared to other species can attribute Paulownia wood a great added value.

In Table 1, the results from the thermogravimetric analysis and elemental composition of 1-, 3-, and 5-year-old Paulownia trees are presented. Overall, the volatile content of the samples that are 1 and 3 years old are similar, while there is a slight increase in the volatile content for the 5-year-old samples. Fixed carbon is similar for the 1- and 3-year-old samples, but it is significantly lower for the 5-year-old samples, meaning that more carbon is lost in the volatile matter. The results presented by Kumar et al. [21], concerning eucalyptus hybrids ranging from 2 to 20 years of age, do not show any significant differences with age in the volatile content. The volatile content is higher than in all the shrubs and small-tree species tested by Nunes et al. [22] in Serra da Estrela. These authors stated that a higher ratio between the volatile content and the fixed carbon leads to a more powerful and rapid combustion. Therefore, 5-year-old trees may cause a more intense and rapid combustion, which could be dangerous, resulting in forest fires. On the other hand, Paulownia trees have been reported to have an extremely high flame retardancy, which, according to Li and Oda [23], is due to the very porous cell tissue of Paulownia wood since the vessels are large and independent, which makes it difficult to ignite. These authors also stated that the thermal conductivity of the carbonized layer is lower than the one for the wood material. The content of ashes decreases with the aging of the samples, which means that there is a lower percentage of inorganic material. Similar results were previously presented for eucalyptus hybrids from 2 to 20 years of age [21]. The results show that 1- and 3-year-old trees fall within the limits imposed by ENPlus® for A2 category wood pellets (1.2%), while 5-year-old trees comply with the limits established by the A1 category wood pellets (0.7%), respectively.

**Table 1.** Thermogravimetric analysis and elemental composition of Paulownia wood that is 1, 3, and 5 years old (n.d.—not detected or presenting values under the detection limit, which is 0.001%).

| Sample | | TGA | | | | CHN | | | S (%) | Cl (%) |
|---|---|---|---|---|---|---|---|---|---|---|
| | | Moisture (%) | Volatile (%) | Ash (%) | Fixed Carbon (%) | C (%) | H (%) | N (%) | | |
| Paulownia (1 year) | 1 | 9.01 | 81.66 | 1.03 | 17.31 | 48.06 | 5.74 | 0.187 | 0.0062 | n.d. |
| | 2 | 8.94 | 80.57 | 0.97 | 18.46 | 48.21 | 5.79 | 0.268 | 0.0064 | n.d. |
| | 3 | 8.89 | 81.82 | 1.00 | 17.18 | 48.38 | 5.82 | 0.251 | 0.0064 | n.d. |
| | Average | 8.94 | 81.35 | 1.00 | 17.65 | 48.22 | 5.78 | 0.235 | 0.0063 | - |
| | Std. Dev. | 0.06 | 0.68 | 0.03 | 0.70 | 0.16 | 0.04 | 0.042 | 0.0002 | - |
| Paulownia (3 years) | 1 | 8.79 | 81.21 | 0.87 | 17.92 | 48.16 | 5.80 | 0.179 | 0.0065 | n.d. |
| | 2 | 8.88 | 81.20 | 0.90 | 17.90 | 48.18 | 5.85 | 0.208 | 0.0066 | n.d. |
| | 3 | 8.87 | 81.33 | 0.81 | 17.86 | 48.10 | 5.86 | 0.213 | 0.0063 | n.d. |
| | Average | 8.85 | 81.25 | 0.86 | 17.89 | 48.14 | 5.83 | 0.200 | 0.0065 | - |
| | Std. Dev. | 0.05 | 0.07 | 0.05 | 0.03 | 0.04 | 0.03 | 0.018 | 0.0002 | - |
| Paulownia (5 years) | 1 | 7.79 | 84.38 | 0.23 | 15.39 | 48.70 | 5.79 | 0.089 | 0.0035 | n.d. |
| | 2 | 7.81 | 84.29 | 0.31 | 15.40 | 48.49 | 5.78 | 0.097 | 0.0036 | n.d. |
| | 3 | 7.77 | 84.54 | 0.28 | 15.18 | 48.47 | 5.82 | 0.077 | 0.0038 | n.d. |
| | Average | 7.79 | 84.40 | 0.28 | 15.32 | 48.55 | 5.80 | 0.087 | 0.0036 | - |
| | Std. Dev. | 0.02 | 0.13 | 0.04 | 0.12 | 0.12 | 0.02 | 0.010 | 0.0001 | - |

Concerning the elemental composition, there seems to be no significant difference between 1- and 3-year-old samples with very similar carbon, hydrogen, nitrogen, and sulfur contents. Even though carbon and hydrogen compounds are similar for the 5-year-old samples, a lower percentage of nitrogen and sulfur was found for these samples. This lower percentage of N shows that 5-year-old wood is more adequate for the production of wood

pellets, since N is responsible for $NO_x$ formation and emission when wood is burnt, which constitutes one of the main environmental impacts of solid biofuel combustion, in parallel to the emission of particulate materials [24]. Nevertheless, all samples fall into the limits of the A1 category wood pellets by ENPlus®, which is 0.3% for nitrogen. Additionally, a small sulfur percentage is essential, since this element contributes to $SO_x$ emissions and is also involved in the formation of aerosols [24]. Similarly, in relation to the sulfur content, the values are much lower than the maximum value for A1 category wood pellets (0.04%). No chloride was found in any of the samples, which is good, since Cl is responsible for deposit formation and corrosion due to HCl formation; this is important for a high plant availability as previously mentioned [24], and, once again, complies with the limits of the ENPlus® standard.

The density and heating values of Paulownia wood of 1, 3, and 5 years old are presented in Table 2. Density is one of the most important wood properties, since it is often associated with several properties, such as mechanical strength, thermal conductivity, or heating value. Paulownia wood that is 1 year old has a very small density, around $0.26 \text{ g} \cdot \text{cm}^{-3}$, which increases with age to $0.42 \text{ g} \cdot \text{cm}^3$ and $0.46 \text{ g} \cdot \text{cm}^3$ for 3-year-old and 5-year-old wood, respectively. However, the wood density for 3- and 5-year-old trees is much higher than that reported for 6-year-old *P. tomentosa* wood grown in Turkey ($0.32 \text{ g} \cdot \text{cm}^{-3}$) [4], or for wood grown in Korea ($0.27 \text{ g} \cdot \text{cm}^{-3}$) [25], and ($0.26 \text{ g} \cdot \text{cm}^3$) [26]. This is possible due to the slower growth rate verified in Portugal, as previously stated [13].

**Table 2.** The density and heating values of Paulownia wood of 1, 3, and 5 years old.

| Sample | | Density (g·cm$^{-3}$) | HHV (MJ·kg$^{-1}$) | LHV (MJ·kg$^{-1}$) |
|---|---|---|---|---|
| Paulownia (1 year) | Average | 0.26 | 19.677 | 18.415 |
| | Std. Dev. | 0.01 | 87.04 | 87.04 |
| Paulownia (3 years) | Average | 0.42 | 19.761 | 18.487 |
| | Std. Dev. | 0.04 | 104.09 | 104.09 |
| Paulownia (5 years) | Average | 0.46 | 19.947 | 18.680 |
| | Std. Dev. | 0.03 | 31.72 | 31.72 |

A slight increase in the Low Heating Value (LHV) and High Heating Value (HHV) of Paulownia wood was observed with aging, from $18.42 \text{ MJ} \cdot \text{kg}^{-1}$ to $18.68 \text{ MJ} \cdot \text{kg}^{-1}$, and $19.68 \text{ MJ} \cdot \text{kg}^{-1}$ to $19.95 \text{ MJ} \cdot \text{kg}^{-1}$, respectively. In accordance to Lachowicz et al. [27], the age of the tree had no significant influence on the heating value of silver birch wood with the ages of 30, 50, and 70 years. Nevertheless, the trees presented in this study are much younger and the wood properties change more in the early years of growth of the trees. A similar increase presented in this study was found for eucalyptus hybrids with ages ranging from 2 to 20 years [21].

Table 3 presents the major elements from Paulownia wood burning with trees aged 1, 3, and 5 years. Potassium and calcium are the two most representative elements, followed by magnesium and phosphorous. Higher levels of Ca, Na, and K have been reported to be responsible for lower deformation temperatures that, according to the ENPlus® standard, have to be higher than 1200 °C for A1 category wood pellets [22]. Nevertheless, the values of these components are much lower than those obtained for several species of shrubs, as presented by Nunes et al. [22]. Potassium decreases with tree age, especially for 1- and 3-year-old trees, compared to 5-year-old trees, showing a higher propensity of 5-year-old trees for wood pellet manufacturing.

Table 4 presents the minor elements from Paulownia wood burning for 1-, 3-, and 5-year-old trees. Concerning the minor elements, no trend was found related to tree aging. Zinc and copper were the most representative compounds, with copper over the maximum limit required by the ENPlus® standard ($<10 \text{ mg} \cdot \text{kg}^{-1}$). Additionally, arsenic was over the limit presented by the standard ($<1 \text{ mg} \cdot \text{kg}^{-1}$). All the other elements were under the limits, similar to the results presented by Nunes et al. [22], for which none of the studied

species complied with the ENPlus® standard limits. The potential variations verified in the content of both major elements, as in the content of minor elements, with the age of the specimens, have been studied by several authors, such as Krutul et al. [28], who analyzed the concentration of some heavy metals in poplar wood, or Szadkowski and Balicka, who analyzed the adsorption of heavy metals by some woods from species, such as European aspen or black locust [29]. However, in none of the situations did the authors present the causes for the variations verified in the content of the different elements, even in the case of other species outside the genus *Paulownia* that could serve as a comparative analysis.

**Table 3.** Major elements from Paulownia wood burning with 1-, 3-, and 5year-old trees (n.d.—not detected or presenting values under the detection limit, which is 0.01 mg·kg$^{-1}$).

| | | Al | Ca | Fe | Mg | P | K | Si | Na | Ti |
|---|---|---|---|---|---|---|---|---|---|---|
| | | | | | **Major Elements (mg·kg$^{-1}$)** | | | | | |
| Paulownia (1 year) | 1 | 10.31 | 1420.58 | 28.72 | 373.14 | 326.07 | 3226.05 | 24.26 | 112.65 | 0.00 |
| | 2 | 12.26 | 1450.10 | 38.46 | 393.00 | 345.75 | 3465.60 | 14.85 | 122.49 | 2.19 |
| | 3 | 12.98 | 1421.55 | 38.79 | 384.80 | 334.45 | 3333.03 | 0.00 | 113.54 | 0.60 |
| | Average | 11.85 | 1430.74 | 35.32 | 383.64 | 335.42 | 3341.56 | 13.04 | 116.23 | 1.40 |
| | Std. Dev. | 1.38 | 16.77 | 5.72 | 9.98 | 9.88 | 120.00 | 12.23 | 5.44 | 1.12 |
| Paulownia (3 years) | 1 | 15.27 | 1366.13 | 70.13 | 482.06 | 338.59 | 2865.36 | 2.96 | 66.00 | 1.62 |
| | 2 | 18.99 | 1588.87 | 119.93 | 498.54 | 323.36 | 2841.04 | 0.00 | 69.62 | 3.89 |
| | 3 | 19.79 | 1600.37 | 85.00 | 506.25 | 317.37 | 2902.64 | 0.00 | 71.01 | 1.87 |
| | Average | 18.02 | 1518.46 | 91.69 | 495.62 | 326.44 | 2869.68 | 2.96 | 68.88 | 2.46 |
| | Std. Dev. | 2.41 | 132.05 | 25.56 | 12.35 | 10.94 | 31.02 | 0.00 | 2.59 | 1.24 |
| Paulownia (5 years) | 1 | 17.91 | 1120.99 | 45.40 | 142.93 | 21.22 | 124.98 | 48.44 | 85.88 | n.d. |
| | 2 | 13.66 | 1027.71 | 54.67 | 126.56 | 23.40 | 104.06 | 45.46 | 76.71 | n.d. |
| | 3 | 17.38 | 1146.65 | 47.49 | 145.16 | 22.47 | 118.88 | 55.76 | 84.74 | n.d. |
| | Average | 16.31 | 1098.45 | 49.19 | 138.21 | 22.37 | 115.97 | 49.89 | 82.44 | - |
| | Std. Dev. | 2.32 | 62.59 | 4.86 | 10.16 | 1.09 | 10.76 | 5.30 | 4.99 | - |

**Table 4.** Minor elements from Paulownia wood burning with 1-, 3-, and 5-year-old trees (n.d.—not detected or presenting values under the detection limit, which is 0.01 mg·kg$^{-1}$).

| | | As | Cd | Co | Cr | Cu | Mn | Ni | Pb | Zn |
|---|---|---|---|---|---|---|---|---|---|---|
| | | | | | **Minor Elements** | | | | | |
| Paulownia (1 year) | 1 | 0.86 | n.d. | 0.00 | n.d. | 20.80 | n.d. | 0.93 | 0.58 | 22.74 |
| | 2 | 1.34 | n.d. | 0.13 | n.d. | 23.01 | n.d. | 0.64 | 0.45 | 26.86 |
| | 3 | 1.14 | n.d. | 0.20 | n.d. | 22.97 | n.d. | 0.45 | 0.00 | 25.06 |
| | Average | 1.11 | - | 0.17 | - | 22.26 | - | 0.67 | 0.34 | 24.89 |
| | Std. Dev. | 0.24 | - | 0.06 | - | 1.26 | - | 0.25 | 0.31 | 2.06 |
| Paulownia (3 years) | 1 | 1.77 | n.d. | 0.51 | n.d. | 8.48 | n.d. | 1.03 | 0.00 | 19.00 |
| | 2 | 1.95 | n.d. | 0.19 | n.d. | 8.13 | n.d. | 0.43 | 0.00 | 17.85 |
| | 3 | 1.42 | n.d. | 0.28 | n.d. | 8.44 | n.d. | 0.52 | 0.00 | 21.04 |
| | Average | 1.72 | - | 0.33 | - | 8.35 | - | 0.66 | 0.00 | 19.30 |
| | Std. Dev. | 0.27 | - | 0.16 | - | 0.19 | - | 0.32 | 0.00 | 1.62 |
| Paulownia (5 years) | 1 | n.d. | 0.02 | 0.10 | n.d. | 16.16 | 1.72 | 0.50 | 0.04 | 9.52 |
| | 2 | n.d. | n.d. | 0.07 | n.d. | 15.05 | n.d. | 0.72 | 0.00 | 8.33 |
| | 3 | n.d. | n.d. | 0.08 | n.d. | 16.87 | n.d. | 0.70 | 0.49 | 9.09 |
| | Average | - | 0.01 | 0.08 | - | 16.03 | 0.57 | 0.64 | 0.18 | 8.98 |
| | Std. Dev. | - | 0.00 | 0.01 | - | 0.92 | 0.00 | 0.12 | 0.32 | 0.60 |

Paulownia wood is known to have low mechanical properties, as previously reported for 6-year-old Paulownia from Turkey [4], with 4280 MPa for MOE and 44 MPa for bending strength, or 3600 MPa for MOE and 42 MPa bending strength for wood grown in

South Korea [25]. The bending strength of 3- and 5-year-old wood, is more or less half of *Pinus pinaster* that had a bending strength of around 108 MPa and 10,900 MPa MOE [30]. No results could be obtained for the 1-year-old samples due to the small size of the trees. Even though the mechanical properties for the 3-year-old samples are higher than the 5-year-old samples, the difference is not significant due to the high standard deviation observed for these mechanical assays (Table 5).

**Table 5.** Mechanical properties of Paulownia wood aged 1, 3, and 5 years.

| Sample | MOE (MPa) | | Bending Strength (MPa) | |
|---|---|---|---|---|
| | Average | Std. Dev. | Average | Std. Dev. |
| Paulownia (3 years) | 6461 | 1098 | 61.7 | 9.8 |
| Paulownia (5 years) | 6990 | 843 | 53.5 | 6.0 |

Figure 2 presents the water absorption of Paulownia wood samples aged 1, 3, and 5 years old. There is a higher water absorption rate for the 1-year-old samples, which reaches more than 100%, compared to the 3- and 5-year-old samples that show absorptions lower than 50%. Regarding the 3- and 5-year-old samples, the differences are minor. Nevertheless, the 3-year-old samples have slightly higher water absorption rates.

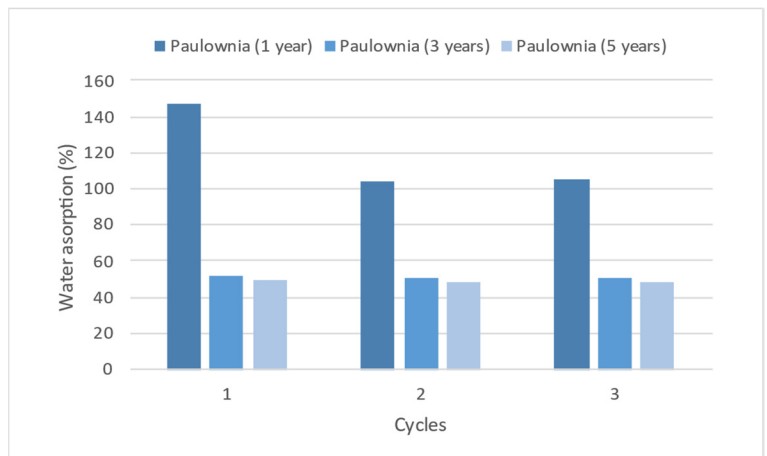

**Figure 2.** Water absorption for 1-, 3-, and 5-year-old Paulownia wood.

Figure 3 presents the dimensional changes between wet and dry environments, along with three cycles. The dimensional changes are compared to the initial dimensions of the samples determined at 0% RH. Generally, the tangential dimensional changes were higher, followed by radial and axial directions. Only the 1-year-old samples showed approximately the same dimensional changes in the radial and tangential directions. This is probably due to the reduced width of the 1-year-old trees, where the radial and tangential directions practically mixed, and therefore pure radial and tangential faces were difficult to attain. This was probably also the reason why dimensional changes were much higher in one-year-old samples that might suffer some distortion after the first wetting cycle. As expected, the dimensional variations in the axial direction were much smaller due to the orientation of the fibers that are mainly oriented in the axial direction, and the water swelling occurred mainly in the transverse directions (radial and tangential). The dimensional changes were much higher in the 1-year-old Paulownia samples, compared to the 3- and 5-year-old samples. This was true for the radial, tangential, and axial directions. This means that 1-year-old Paulownia wood has a very low dimensional stability, not permitting its use as a solid wood. Nevertheless, due to the characteristic hole in this type of wood, it would also be challenging to obtain a usable board from 1-year-old samples. Concerning the 3- and 5-year-old samples, their difference is relatively small. However, the dimensional changes seem to be higher for the 3-year-old samples. There were no major differences in

the dimensional chances in the first three cycles, which leads us to conclude that there was no significant deterioration of the sample after 3 cycles of exposure to dry (100 °C) and wet (immersion in water) steps.

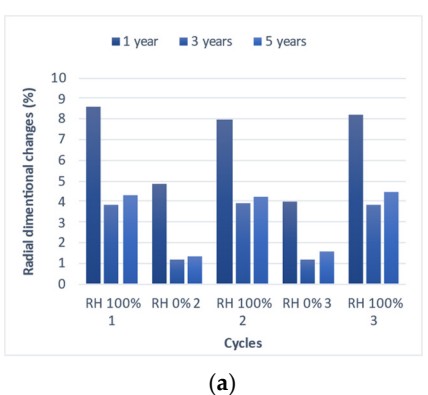

(a)

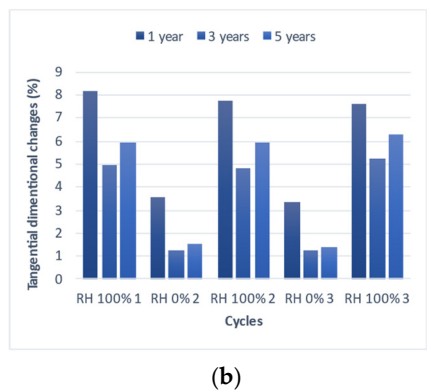

(b)

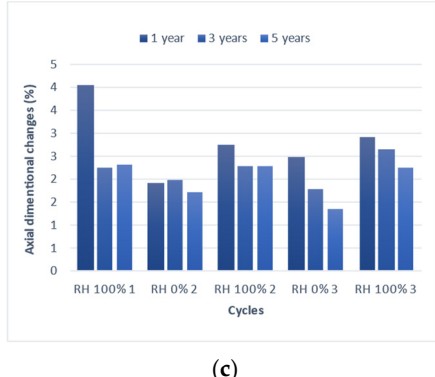

(c)

**Figure 3.** Dimensional changes: (**a**) radial direction; (**b**) tangential direction; and (**c**) axial direction.

## 4. Conclusions

This work intended to determine the properties of wood from very young *Paulownia tomentosa* trees grown in Portugal to find the best valorization path. Generally, the results show that, chemically there is an increase in the extractive content with the age of the tree, and this increase is mainly due to the ethanol and water extracts. Nevertheless, no substantial differences were found for lignin or α-cellulose, but hemicelluloses decreased with age. Overall, the volatile content of the samples aged 1 and 3 years old are similar, while there is a slight increase in the volatile content of the 5-year-old samples. Fixed carbon is similar for the 1-year-old and 3-year-old samples, but it is significantly lower for the 5-year-old samples. Therefore, more carbon is lost in the volatile matter, and combustions are more powerful and rapid. The content of ashes decreases along with the age of the samples, which means that there is a lower percentage of inorganic material. Additionally, the elemental composition is similar for 1-year-old and 3-year-old samples, but better for 5-year-old samples, namely, because of the lower percentages of S and N, which shows that 5-year-old wood is adequate for pellet production. Potassium and calcium are the two most representative elements, followed by magnesium and phosphorous. Some minor elements, such as copper and arsenic, were over the maximum limit required by the ENPlus® standard. A slight increase in the Low Heating Value (LHV) and High Heating Value (HHV) was also observed with tree age. The density of 1-year-old Paulownia wood is minimal and increases along with the tree age. Although the bending strength and MOE are low, they are much higher than Paulownia growth in other regions. The dimensional changes were much higher in 1-year-old Paulownia samples, compared to the 3- and 5-year-old samples. Therefore, the age of 5 years is the minimum cutting age for the use of this wood for solid fuel, such as pellets, and as solid wood.

**Author Contributions:** Conceptualization, B.E.; methodology, B.E., L.J.R.N., I.D. and L.C.-L.; formal analysis, B.E., I.D., L.C.-L. and L.J.R.N.; investigation, H.V., B.E., I.D. and L.C.-L.; resources, H.V. and J.F.; writing—original draft preparation, B.E.; writing—review and editing, B.E., L.J.R.N., I.D., H.V., L.C.-L. and J.F.; project administration, B.E.; funding acquisition, B.E., L.J.R.N., I.D., H.V., L.C.-L. and J.F. All authors have read and agreed to the published version of the manuscript.

**Funding:** This work was conducted in the framework of Project VALPT (PROJ/IPV/ID&I/003) financed by Caixa Geral de Depósitos and also financed by national funds through FCT—Fundação para a Ciência e Tecnologia, I.P., through the CERNAS Research Centre, within the scope of the project UIDB/00681/2020. L.J.R.N. was supported by proMetheus, Research Unit on Energy, Materials and Environment for Sustainability—UIDP/05975/2020, funded by national funds through FCT—Fundação para a Ciência e Tecnologia and H.V supported by CITAB, Centre for the Research and

Technology of Agro-Environmental and Biological Sciences—Portuguese Foundation for Science and Technology—Project UIDB/04033/2020.

**Institutional Review Board Statement:** Not applicable.

**Informed Consent Statement:** Not applicable.

**Data Availability Statement:** Data are available on request from the corresponding author.

**Conflicts of Interest:** The authors declare no conflict of interest.

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
