# Peer review of "The Influence of Age on the Wood Properties of Paulownia tomentosa (Thunb.) Steud."

_forests, doi:10.3390/f13050700_

Round 1

Reviewer 1 Report

This article presents an interesting niche topic in woodworking issues. The issues raised in the paper may gain in importance due to the decreasing age of felling trees caused by the high demand for this material. However, the authors did not avoid several serious, in my opinion, shortcomings that should be corrected in the paper. 

The methodology presents the determination of alpha-cellulose content in holocellulose. However, in the Results and Discussion (Section 3, Table 1), this is presented as the cellulose content of the test samples. This is an error. Because the method itself determines the alpha-cellulose content after previous dissolution of the hemicelluloses, but also beta-cellulose and gamma-cellulose (beta-cellulose can then be determined in the filtrate by digestion).  

For a correct determination of cellulose in the analysed material it was possible to use the Kürschner-Hoffer or Seifert method (for the determination of pentosans by means of the difference in holocellulose and cellulose content it is more accurate to use the Kürschner-Hoffer method because less beta-cellulose is lost in this determination (the gamma cellulose is dissolved in each determination)). 

https://doi.org/10.1007/BF02626309

https://doi.org/10.1515/hfsg.1962.16.1.1

Why were polar haraktee solvents used for extraction and not a mixture of solvents (polar-nonpolar). The results of the determination of extraction substances do not take into account extraction substances of non-polar nature. 

DOI:10.15199/62.2016.11.5

https://doi.org/10.3390/f12050647

Table 1. shows the values of 1 measurement or the average of the measurements. The standard deviation is also missing there. The results would be better presented as a graph in my opinion. In the table, the name cellulose should be changed to alpha-cellulose because of the determination performed. 

The results presented in Table 3 and . are not discussed/commented on in the text. No standard deviations in Figures 1 and 2.

The values of the mineral compounds presented in Table 5 are, in my opinion, incorrectly presented. If in 2 measurements out of 3 the content of a given element was not detected and in one it was determined (Cr 0.24, Mn 1.72, Cd 0.02) or one measurement was significantly higher than the others ( Ni 1.03/0.43/0.52 Pb 0.49/0.00/0.04) it suggests more that the outlier measurements are erroneous (e.g. caused by sample or instrument contamination). The value of 0.00 alone is not a true value as the quantification of a substance may be at the limit of quantification for a given method (hence it is more precise to replace 0.00 by not detected - N/D). 

No discussion of what may influence the content of particular compounds in the analysed wood. And why it changes with age. The following articles may help in this respect although they refer to a different tree species.  

DOI:10.37763/wr.1336-4561/66.1.3948

DOI:10.5604/01.3001.0014.6773

DOI:10.37763/wr.1336-4561/66.1.3948

There is a lack of statistical elaboration of the presented results (test of significance of differences e.g. ANOVA test).  

Author Response

Comments from Reviewer #1:

This article presents an interesting niche topic in woodworking issues. The issues raised in the paper may gain in importance due to the decreasing age of felling trees caused by the high demand for this material. However, the authors did not avoid several serious, in my opinion, shortcomings that should be corrected in the paper. 

Answer to comments from Reviewer #1: The authors would like to thank Reviewer #1 for the time spent reading and commenting the article and for all the constructive comments presented. The authors tried to address all the comments presented as described in the following listed point by point explanations. The authors would like as well to thank Reviewer #1 for considering that this article presents as interesting niche topic in woodworking issues.

The methodology presents the determination of alpha-cellulose content in holocellulose. However, in the Results and Discussion (Section 3, Table 1), this is presented as the cellulose content of the test samples. This is an error. Because the method itself determines the alpha-cellulose content after previous dissolution of the hemicelluloses, but also beta-cellulose and gamma-cellulose (beta-cellulose can then be determined in the filtrate by digestion).  For a correct determination of cellulose in the analysed material it was possible to use the Kürschner-Hoffer or Seifert method (for the determination of pentosans by means of the difference in holocellulose and cellulose content it is more accurate to use the Kürschner-Hoffer method because less beta-cellulose is lost in this determination (the gamma cellulose is dissolved in each determination)). 

Answer to comments from Reviewer #1: Indeed we have determined several times the Kürschner-Hoffer cellulose content before but in this case α-cellulose was determined, therefore the term was corrected throughout the text. For other fast growing trees, such as eucalyptus and other Paulownia species, α-cellulose was also the method chosen which allowed us to make some comparisons to other materials.

(Ashori & Nourbakhsh, 2009; Carrillo et al., 2018; Ramirez et al., 2009; Welter, 2021)

Why were polar haraktee solvents used for extraction and not a mixture of solvents (polar-nonpolar). The results of the determination of extraction substances do not take into account extraction substances of non-polar nature. 

Answer to comments from Reviewer #1: The use of three different solvents allows the determination of non-polar extractives removed by (Dichloromethane), semi-polar (ethanol) and polar (water), similar to the mentioned in the article (water, ethanol, cyclohexane) referred by the reviewer that was also cited in the text.

DOI:10.15199/62.2016.11.5

https://doi.org/10.3390/f12050647

Table 1. shows the values of 1 measurement or the average of the measurements. The standard deviation is also missing there. The results would be better presented as a graph in my opinion. In the table, the name cellulose should be changed to alpha-cellulose because of the determination performed. 

Answer to comments from Reviewer #1: The average of the measurements. The changes were made accordingly to the suggested. Standard deviations are presented as error bars in the Figure.

The results presented in Table 3 and . are not discussed/commented on in the text. No standard deviations in Figures 1 and 2.

Answer to comments from Reviewer #1: The table was not mentioned in the text but that mistake has been corrected.

The values of the mineral compounds presented in Table 5 are, in my opinion, incorrectly presented. If in 2 measurements out of 3 the content of a given element was not detected and in one it was determined (Cr 0.24, Mn 1.72, Cd 0.02) or one measurement was significantly higher than the others ( Ni 1.03/0.43/0.52 Pb 0.49/0.00/0.04) it suggests more that the outlier measurements are erroneous (e.g. caused by sample or instrument contamination). The value of 0.00 alone is not a true value as the quantification of a substance may be at the limit of quantification for a given method (hence it is more precise to replace 0.00 by not detected - N/D). 

Answer to comments from Reviewer #1: Thank you very much for this comment. The authors entirely agree with Reviewer #1. In fact, where was presented the value zero, it was intended to present the value n.d. (not detected). The situation was corrected as suggested by Reviewer #1.

No discussion of what may influence the content of particular compounds in the analysed wood. And why it changes with age. The following articles may help in this respect although they refer to a different tree species.  

DOI:10.37763/wr.1336-4561/66.1.3948

DOI:10.5604/01.3001.0014.6773

DOI:10.37763/wr.1336-4561/66.1.3948

Answer to comments from Reviewer #1: Thank you very much for this comment. The authors entirely agree with Reviewer #1 and included the missing discussion. The references (2) were analyzed and cited as suggested by Reviewer #1.

There is a lack of statistical elaboration of the presented results (test of significance of differences e.g. ANOVA test).  

Answer to comments from Reviewer #1: Thank you very much for this comment. The authors understand the recommendation presented by Reviewer #1. However, if the Reviewer #1 do not mind, the authors would like to present a different opinion in this occasion. The authors consider that in this precise case the statistical analysis do not present any new achievement to the text, because the main objective is to see if the results obtained fit comparatively with the limit values presented in the standard ENPlus ®. So, in this case, the authors kindly ask Reviewer #1 to accept the explanation.

The authors would like to thank once again to Reviewer #1 for the time spent reading and commenting the article, and for the comments and recommendations presented. The authors tried to address all of them the best way possible and expect that now Reviewer #1 can accept the article as it is. Thank you very much.

Reviewer 2 Report

Lines 55-56: what does it mean that “species has a specific mass of approximately 35 g.cm-3”. That means its density is equal to 3500 kg/m3. In reviewers opinion it should be 0.35 g x cm-3.

 At the end of the introduction to the article, there is no clearly defined aim of the paper.

How many trees were tested on?

In the Materials and Methods chapter, you can consider shortening the description of the methodology for tests performed according to standards.

The sample preparation flowchart would be very helpful. I am particularly interested in the method of preparing samples from one-year-old wood. Was the wood layer, around the pith, sufficient to obtain 20x20 mm samples? Did the 3 and 5-year-old wood samples contain only 3rd and 5th annual growth rings or all growth rings? What was the diameter of the trees from which 1, 3 and 5-year-old samples were obtained?

Please explain why in the bend test the 340 mm distance between the supports was used? For samples with a height of 20 mm, the most common distance between supports is 240 mm, which is 12 times greater than the height of the sample. What is the force of 735.5 N and where did it come from? Is it a calculated value representing some percentage of the destructive force?

Formula 7: what is 9.8 in the formula, you used it also in next one.

MPa unit not Mpa, ex formula 8.

Lines 191-192: Three samples were used to bend one-year-old wood. Was it not possible to multiply the number of samples by cutting down additional trees?

Lines 193-194: treated and untreated wood? Did you use some wood modification?

Line 214: unfinished sentence.

Table 1: replace decimal separator from commas to period. Are the presented differences statistically significant or not? How do you explain the sharp decline in lignin and hemicelluloses in five-year-old wood?

Lines 249-265: why, suddenly, are you giving values for pellets even though you haven't talked anywhere about using this wood for pellets before? What is “ENplus ® for Category A2 pellets”? Is this the standard? Is this an important message for the article?

Table 2: relative humidity or moisture content?

 Lines 312-317: How is it possible that such light wood has MOE and MOR on the level of mahogany or even ipe ?? MOE is a measure of stiffness and, as you have given yourself, wood had high flexibility. From the point of view of wood science, such a result is impossible for juvenile wood. Did the sample contain a tree core? In Paulownia, they are often empty and of large diameter. The presence of the core could distort the measurement.

Line 320: however, were statistical analyzes performed? The differences are said to not be statistically significant.

Figure 1: Surprising differences between samples. How do you explain that? Did the sample from 3-year-old Paulownia contain growth rings no. 1,2,3 or only the third? Since the first annual growth has adsorbed 1.5 times more water than it weighs, why the rings 1,2,3 are only 50%?

Figure 2: I do not understand the value of e.g. 5%, in the radial direction of the one-year sample for 0% RH. Is it a shrinkage grade value? Against which starting dimension the changes were calculated?

Author Response

Comments from Reviewer #2:

Answer to comments from Reviewer #2: The authors would like to thank Reviewer #2 for the time spent reading and commenting the article and for all the constructive comments presented. The authors tried to address all the comments presented as described in the following listed point by point explanations.

Lines 55-56: what does it mean that “species has a specific mass of approximately 35 g.cm-3”. That means its density is equal to 3500 kg/m3. In reviewers opinion it should be 0.35 g x cm-3.

Answer to comments from Reviewer #2: Thank you very much for this comment. The authors entirely agree with Reviewer #2 and corrected the situation as suggested by Reviewer #2.

 At the end of the introduction to the article, there is no clearly defined aim of the paper.

Answer to comments from Reviewer #2: Thank you very much for this comment. The corrected the situation as suggested by Reviewer #2 and included a paragraph with the objectives of the article.

How many trees were tested on?

Answer to comments from Reviewer #2: The number of trees was dependent on the age of trees. For example, for 1 year paulownia trees 6 different trees (from the same plantation) were needed to cut all the samples while for 3 and 5 years old only two trees were needed. This information was added in the material and methods section.

In the Materials and Methods chapter, you can consider shortening the description of the methodology for tests performed according to standards.

Answer to comments from Reviewer #2: Thank you very much for this comment. The situation was addressed as suggested by Reviewer #2 and the section was shortened in accordance.

The sample preparation flowchart would be very helpful. I am particularly interested in the method of preparing samples from one-year-old wood. Was the wood layer, around the pith, sufficient to obtain 20x20 mm samples? Did the 3 and 5-year-old wood samples contain only 3rd and 5th annual growth rings or all growth rings? What was the diameter of the trees from which 1, 3 and 5-year-old samples were obtained?

Answer to comments from Reviewer #2: Part of the answer was already given before. 6 different trees from 1 year old were needed to obtain the 20x20 samples and only 3 well oriented samples could be obtained. The samples with 3 and 5 years old contained all the growth rings, therefore only two trees were enough to obtain the samples. The diameter of the trees was variable. This information was added in the material and methods section.

Please explain why in the bend test the 340 mm distance between the supports was used? For samples with a height of 20 mm, the most common distance between supports is 240 mm, which is 12 times greater than the height of the sample. What is the force of 735.5 N and where did it come from? Is it a calculated value representing some percentage of the destructive force?

Formula 7: what is 9.8 in the formula, you used it also in next one.

Answer to comments from Reviewer #2: Thank you very much for this comment. The span used was in accordance to the Portuguese standard NP-619 that states that the span should be between 280 to 320 mm. In fact there was an error because the span used was 300 mm which is the necessary for the 360 mm samples. The force 735.5 N is 10% of the average maximum force obtained in the bending strength assays. This information was included in the text. The 9.8 value is used to convert from kgf to N.

MPa unit not Mpa, ex formula 8.

Answer to comments from Reviewer #2: Thank you very much for this comment. The situation was corrected in accordance with the suggestion presented by Reviewer #2 and all the entire document was checked to eliminate any other similar errors.

Lines 191-192: Three samples were used to bend one-year-old wood. Was it not possible to multiply the number of samples by cutting down additional trees?

Answer to comments from Reviewer #2: Thank you very much for this comment. Unfortunately, the authors were not allowed to cut more trees to proceed with additional tests. It would not be possible to obtain good oriented specimens from most of the trees. 6 good trees were necessary to obtain the necessary specimens. The authors kindly ask Reviewer #2 to accept this explanation.

Lines 193-194: treated and untreated wood? Did you use some wood modification?

Answer to comments from Reviewer #2: Thank you very much for this comment. The use of the expression treated and untreated wood was wrongly used and gave a wrong sense to the sentence in the translation process. The situation was now corrected and the words were removed.

Line 214: unfinished sentence.

Answer to comments from Reviewer #2: Thank you very much for this comment. The situation was corrected in accordance with the suggestion presented by Reviewer #2.

Table 1: replace decimal separator from commas to period. Are the presented differences statistically significant or not? How do you explain the sharp decline in lignin and hemicelluloses in five-year-old wood?

Answer to comments from Reviewer #2: Thank you very much for this comment. The authors proceeded to the replacement of all commas by dots as suggested by Reviewer #2. Concerning lignin content, the increase verified between the 1st and 3rd years was followed by a decreased in the 5 years-old samples, but the differences were relatively small. The same was observed by Domingos et al. with eucalyptus wood, in which the percentage of Klason Lignin was approximately constant between 6-15 years of age, ranging from 19.6% to 23.1%. On the other hand, according to Miranda and Pereira, there was an increase in the lignin content of Eucalyptus globulus trees with 2, 3 and 6 years old. The same was found concerning soluble lignin content, which varied between 3.3% and 4.0%. Similar results were reported for Eucalyptus grandis wood with four different ages (10, 14, 20 and 25 years). The explanation was added to the text as requested by Reviewer #2.

Lines 249-265: why, suddenly, are you giving values for pellets even though you haven't talked anywhere about using this wood for pellets before? What is “ENplus ® for Category A2 pellets”? Is this the standard? Is this an important message for the article?

Answer to comments from Reviewer #2: Thank you very much for this comment. The reference to the ENPlus ® standard was introduced in the Materials and Methods section. The reference is used just to have a comparative analysis with well-known data as those presented in ENPlus.

Table 2: relative humidity or moisture content?

Answer to comments from Reviewer #2: Thank you very much for this comment. It is moisture content. The situation was corrected in accordance with the suggestion presented by Reviewer #2.

Lines 312-317: How is it possible that such light wood has MOE and MOR on the level of mahogany or even ipe ?? MOE is a measure of stiffness and, as you have given yourself, wood had high flexibility. From the point of view of wood science, such a result is impossible for juvenile wood. Did the sample contain a tree core? In Paulownia, they are often empty and of large diameter. The presence of the core could distort the measurement.

Answer to comments from Reviewer #2: No core was present in the samples. The results presented are a measure of bending strength determined by a three point bending test and accordingly to a Portuguese standard. The results are very similar to all the other results presented in the literature for different Paulownia species and for the same species from different proveniences as mentioned in the text (Akyildiz & Kol, 2010; Hidayat et al., 2017; Kim et al., 2018). MOR is usually determined by a four point bending test and results are therefore different. Some authors incorrectly call it MOR but this test should be called bending strength alone.

Line 320: however, were statistical analyzes performed? The differences are said to not be statistically significant.

Answer to comments from Reviewer #2: Thank you very much for this comment. Concerning the use of the word “significantly” it was wrongly translated and was not used in the appropriate mode. The authors rearranged the sentence to clarify the correct sense and to avoid misunderstandings to the readers.

Figure 1: Surprising differences between samples. How do you explain that? Did the sample from 3-year-old Paulownia contain growth rings no. 1,2,3 or only the third? Since the first annual growth has adsorbed 1.5 times more water than it weighs, why the rings 1,2,3 are only 50%?

Answer to comments from Reviewer #2: Thank you very much for this comment. The samples were a mixture of all growth rings. The reason is exactly what the referee stated. There is a huge difference between 1 year Paulownia wood and 3 years old as the results show, that was the objective of this study.

Figure 2: I do not understand the value of e.g. 5%, in the radial direction of the one-year sample for 0% RH. Is it a shrinkage grade value? Against which starting dimension the changes were calculated?

Answer to comments from Reviewer #2: Thank you very much for this comment. These results are the differences between the initial 0% RH and after 1, 2 and 3 cycles of 100% RH, followed by 0% RH. This was better explained in the text.

The authors would like to thank once again to Reviewer #2 for the time spent reading and commenting the article, and for the comments and recommendations presented. The authors tried to address all of them the best way possible and expect that now Reviewer #2 can accept the article as it is. Thank you very much.

Akyildiz, M. H., & Kol, H. S. (2010). Some technological properties and uses of paulownia (Paulownia tomentosa Steud.) wood. Journal of Environmental Biology, 31(3), 351–355.

Ashori, A., & Nourbakhsh, A. (2009). Studies on Iranian cultivated paulownia–a potential source of fibrous raw material for paper industry. European Journal of Wood and Wood Products, 67(3), 323–327. https://doi.org/10.1007/s00107-009-0326-0

Carrillo, I., Mendonça, R. T., Ago, M., & Rojas, O. J. (2018). Comparative study of cellulosic components isolated from different Eucalyptus species. Cellulose, 25(2), 1011–1029. https://doi.org/10.1007/s10570-018-1653-2

Hidayat, W., Qi, Y., Jang, J.-H., Febrianto, F., & Kim, N. H. (2017). Effect of Mechanical Restraint on the Properties of Heat-treated Pinus koraiensis and Paulownia tomentosa Woods. BioResources, 12(4), 7539–7551.

Kim, Y. K., Kwon, G. J., Kim, A. R., Lee, H. S., Purusatama, B., Lee, S. H., Kang, C. W., & Kim, N. H. (2018). Effects of heat treatment on the characteristics of royal paulownia (Paulownia tomentosa (Thunb.) Steud.) wood grown in Korea. Journal of the Korean Wood Science and Technology, 46(5), 511–526. https://doi.org/10.5658/WOOD.2018.46.5.511

Ramirez, M., Rodriguez, J., Balocchi, C., Peredo, M., Elissetche, J. P., Mendonça, R., & Valenzuela, S. (2009). Chemical composition and wood anatomy of Eucalyptus globulus clones: Variations and relationships with pulpability and handsheet properties. Journal of Wood Chemistry and Technology, 29(1), 43–58. https://doi.org/10.1080/02773810802607559

Welter, C. A. (2021). Bioprodutos obtidos da madeira de Paulownia tomentosa Steud. [PhD Thesis]. Universidade Federal de Santa Maria.

Round 2

Reviewer 1 Report

I thank the authors for addressing my comments on the article. However, I would still wonder about the results presented in e.g. Table 4 due to the fact that the mean of the measurements of e.g. Cr (0.24, n.d, n.d) is not 0.08 with a standard deviation of 0.00 in my opinion more suggestive of sample contamination where the element was detected. 

Table 5 Paulownia 1 year MOE 17.993 with a standard deviation of 2037? Additionally, the comma needs to be replaced by a full stop. 

However, I consider these errors to be editorial mistakes and would ask the authors to review the manuscript in this respect and make the necessary corrections. 
Thank you again for taking my suggestions into consideration and I wish you further such interesting work. 

Author Response

Reviewer 1

I thank the authors for addressing my comments on the article. However, I would still wonder about the results presented in e.g. Table 4 due to the fact that the mean of the measurements of e.g. Cr (0.24, n.d, n.d) is not 0.08 with a standard deviation of 0.00 in my opinion more suggestive of sample contamination where the element was detected. 

Answer to comments from Reviewer #1: The authors would like to thank Reviewer #1 for this comment. The authors entirely agree with Reviewer #1 and corrected the situation in accordance with the suggestion presented by Reviewer #1. The data was removed and corrected as being a contamination and excluded for being non-significant for the results.

Table 5 Paulownia 1 year MOE 17.993 with a standard deviation of 2037? Additionally, the comma needs to be replaced by a full stop. 

Answer to comments from Reviewer #1: Thank you for your comment. The one-year-old samples  were removed from the static bending tests due to some difficulty in obtaining well-oriented samples.

However, I consider these errors to be editorial mistakes and would ask the authors to review the manuscript in this respect and make the necessary corrections. 
Thank you again for taking my suggestions into consideration and I wish you further such interesting work. 

Answer to comments from Reviewer #1: The authors would like to thank Reviewer #1 for the time spent reading and commenting the article.

Reviewer 2 Report

Lines 319-328: The modulus of elasticity is not a measure of the flexibility but the stiffness of a material. The higher the MOE, the material transmits greater stresses with smaller deformations. The one-year Paulownia tree is not stiff and I am almost sure that the deflection in the bend test was much greater than for the other samples. The MOE value of 17 GPa is unbelievable for wood with a density of 260 kg / m3. Please explain how the deflection of the sample was measured during the test. I understand that in the formula for MOE, delta X means the sample deflection? According to wood database, Wagenfuhr and similar, the modulus of elasticity of Paulownia wood is approximately 4300 MPa and the bending strength is approximately 40 MPa. You even quote in the text that MOE is at a similar level. Please look carefully at the deflection of the samples during the test. In the swelling section you wrote that the one-year samples had similar swelling in the radial and tangential directions because it was difficult to cut an oriented sample. So, in the bending test, were the samples well-oriented and were the force acting in the direction tangent to the annual ring?

Figure 2: how do you explain such large differences in water absorption? The size of the anatomical elements and their proportion do not change so much within the three annual growth rings. Does the value of 5% for RH 0% mean that the sample in the dry state was as much as 5% larger after the first humidification cycle in relation to the initial size? Was it 5% higher after the next cycle? I don't understand these values in the dry samples.

Author Response

Rewiewer 2

Lines 319-328: The modulus of elasticity is not a measure of the flexibility but the stiffness of a material. The higher the MOE, the material transmits greater stresses with smaller deformations. The one-year Paulownia tree is not stiff and I am almost sure that the deflection in the bend test was much greater than for the other samples. The MOE value of 17 GPa is unbelievable for wood with a density of 260 kg / m3. Please explain how the deflection of the sample was measured during the test. I understand that in the formula for MOE, delta X means the sample deflection? According to wood database, Wagenfuhr and similar, the modulus of elasticity of Paulownia wood is approximately 4300 MPa and the bending strength is approximately 40 MPa. You even quote in the text that MOE is at a similar level. Please look carefully at the deflection of the samples during the test. In the swelling section you wrote that the one-year samples had similar swelling in the radial and tangential directions because it was difficult to cut an oriented sample. So, in the bending test, were the samples well-oriented and were the force acting in the direction tangent to the annual ring?

Answer to comments from Reviewer #2: The authors would like to thank Reviewer #2 for the time spent reading and commenting the article. Indeed, it was very difficult to cut specimens for the bending strength and stiffness since the one-year Paulownia trees did not grow enough and we cannot guarantee that the specimens were well oriented and that the direction was tangent to the annual ring. We could only cut three samples with 360x20x20 dimensions and they were not “perfectly well oriented”. Therefore, we decided to exclude the 1-year samples from the static bending tests. For the remaining tests it was possible to obtain good samples because they were much smaller.

Figure 2: how do you explain such large differences in water absorption? The size of the anatomical elements and their proportion do not change so much within the three annual growth rings. Does the value of 5% for RH 0% mean that the sample in the dry state was as much as 5% larger after the first humidification cycle in relation to the initial size? Was it 5% higher after the next cycle? I don't understand these values in the dry samples.

Answer to comments from Reviewer #2: As mentioned in the text in lines 356 and 357 the dimensional changes are compared to the initial dimensions of the samples determined at 0% RH. The 5%-dimensional changes in the dry state after the 1st cycle means that the sample increased its dimension in 5%(for example from 20.35 mm to 21.28 mm) after an immersion step followed by drying until 0% RH. It is normal to samples suffer some distortion after the wet and dry cycles. That happened specially in 1-year-old samples which the authors acknowledge that might be due to some poor oriented samples. This was added in the discussion.